# Using Soluble ST2 to Predict Adverse Postoperative Outcomes in Patients with Impaired Left Ventricular Function Undergoing Coronary Bypass Surgery [note 1]

**DOI:** 10.3390/medicina55090572

**Published:** 2019-09-07

**Authors:** Ahmet Dolapoglu, Eyup Avci, Tarik Yildirim, Hasan Kadi, Ahmet Celik

**Affiliations:** 1Department of Cardiovascular Surgery, Balikesir University Medical School, 10145 Balikesir, Turkey; 2Department of Cardiology, Balikesir University Medical School, 10145 Balikesir, Turkey (E.A.) (H.K.); 3Department of Cardiology, Mugla University Medical School, 48000 Mugla, Turkey; 4Department of Cardiology, Mersin University Medical School, 33000 Mersin, Turkey

**Keywords:** soluble ST2, coronary artery bypass graft surgery, postoperative adverse events

## Abstract

*Background and Objectives:* The aim of this study was to investigate the prognostic value of soluble ST2 (sST2) in predicting postoperative adverse events in patients with impaired left ventricular (LV) function undergoing coronary artery bypass graft (CABG) surgery. *Materials and Methods:* This study included 80 consecutive patients with stable coronary artery disease (CAD) and impaired LV function (ejection fraction ≤ 45%) undergoing on-pump coronary artery bypass graft surgery. The patients were divided into the “high” or “low” group according to their ST2 levels (≥35 or <35 ng/mL). *Results:* Postoperative adverse events were more common in patients with high sST2 levels than in patients with low sST2 levels (100% vs 26%, *p* < 0.0001). Multivariate analysis showed that sST2 level was an independent predictor of the presence of postoperative adverse events (OR: 1.117 (95% CI: 1.016–1.228), *p* = 0.022). The receiver operating characteristic curve (ROC) analysis of sST2 revealed an area under the curve (AUC) of 0.812 (95% CI: 0.710–0.913, *p* < 0.001) in predicting postoperative adverse events. An sST2 level of 26.50 ng/ml was identified as the optimal cut-off value, with a sensitivity and specificity of 74.1% and 75.3%, respectively. *Conclusion:* Higher sST2 levels were associated with adverse outcomes after CABG in patients with impaired LV and stable CAD.

## 1. Introduction

Coronary artery bypass graft (CABG) surgery is a good treatment option for patients with stable coronary artery disease [1]. However, the main arguments against this surgery are postoperative mortality and complications in patients with impaired left ventricular function. Many biomarkers have previously been used to predict adverse outcomes after CABG surgery, such as creatine kinase (CK)-MB isoenzyme, cardiac troponin, N-terminal pro-brain natriuretic peptide (NT-proBNP), and neutrophil gelatinase-associated lipocalin [2].

The ST2 gene encodes a protein that is a member of the interleukin-1 receptor family. This protein consists of two isoforms: a transmembrane form and a soluble form. Serum levels of soluble ST2 (sST2) markedly increase during myocardial stretch. Furthermore, there is a well established association between elevated sST2 levels and impaired left ventricular function [3]. Previous studies have shown that serum levels of sST2 increase significantly after acute myocardial infarction (MI) and acute decompensated heart failure [4]. In clinical settings, sST2 level is predominantly used to predict mortality after early acute MI or acutely decompensated heart failure since the sST2 baseline is elevated due to myocardial stretch caused by ventricular volume overload [5,6]. Moreover, in studies of postoperative outcomes after cardiac surgery, elevated sST2 levels were found to be associated with increased mortality and kidney failure [7,8]. However, the patients in these studies primarily had normal ventricular function. Impaired left ventricular function has been cited as a risk factor for mortality and complications after cardiac surgery [9].

The relationship between preoperative sST2 levels and postoperative outcomes has not previously been investigated in patients with stable coronary artery disease (CAD) and impaired left ventricular function who undergo CABG. Therefore, the purpose of this study was to assess the use of sST2 levels to predict postoperative adverse events in such patients.

## 2. Materials and Methods

### 2.1. Design

This study included 80 patients, including 19 women, with impaired left ventricular function (ejection fraction ≤45%) who underwent isolated on-pump CABG between February 2016 and January 2017. Demographic and clinical characteristics, surgical details, and postoperative outcomes were obtained from the patients’ charts. We excluded patients with acute coronary syndrome, as well as those with elevated troponin levels, acute or chronic kidney disease, severe chronic obstructive pulmonary disease, abnormal cardiac rhythm, valvular heart disease (including moderate or severe regurgitation or severe stenosis), previous cardiac surgery, and acute MI-related mechanical complications, such as left ventricular wall rupture, a ventricular septal defect, or papillary muscle rupture.

### 2.2. Ethics

The study complies with the Declaration of Helsinki and was approved by our institutional ethics committee (Ethical approval number 2016/123, approved on 21.12.2016). Written informed consent was obtained from all patients.

### 2.3. Blood Sampling

Blood samples were obtained from the patients before surgery for routine hemogram analysis and the analysis of biochemistry panels. The serum used for measurements of sST2 and NT-proBNP was isolated within 1 hour of collection, frozen at –20 °C, immediately stored at −80 °C, and then shipped in dry ice to the laboratory. ST2 was measured with an immunoassay (Medical and Biological Labs, Woburn, MA, USA) and NT-proBNP was measured with an Elecsys 2010 immunoassay analyzer (Roche Diagnostics, Indianapolis, IN, USA). The baseline sST2 level of >35 ng/ml has been accepted as a predictor of adverse outcomes [10].

### 2.4. Definitions

The diagnosis of CAD requiring surgical revascularization was confirmed by coronary angiography. Echocardiographic examinations were performed in all patients prior to CABG surgery via transthoracic echocardiography. Values of left ventricular ejection fraction (LVEF) were calculated after measuring the end-diastolic and end-systolic left ventricle volumes in the apical four-chamber and two-chamber views using the modified Simpson’s method. The decision to perform CABG was made by the local “heart team,” which consists of cardiologists and cardiovascular surgeons. We collected data related to the patients’ clinical and demographic characteristics, including age, gender, body mass index (BMI), LVEF, left ventricular end-systolic diameter (LVESD), left ventricular end-diastolic diameter (LVEDD), creatinine and blood urea nitrogen (BUN) levels, syntax score, surgical risk score (STS score), New York Heart Association (NYHA) classification, left main coronary artery (LMCA) disease, number of the coronary vessel disease, serum sST2 level, and NT-proBNP level. We also collected data related to comorbidities, including diabetes mellitus (DM), hypertension (HT), hyperlipidemia, smoking, chronic pulmonary obstructive disease (COPD), history of previous MI, percutaneous coronary intervention (PCI), cerebrovascular disease (CVD), and peripheric arterial disease (PAD), as well as surgical details, including those related to the left internal mammarian artery (LIMA), the number of by-passed vessels, whether endarterectomy was performed, the cardiopulmonary bypass (CPB) time, and the cross-clamp (X-Clamp) time. The presence of any of the following events was defined as postoperative adverse events: all causes of death, newly developed atrial fibrillation, post-operative requirement of inotropic agents or intra-aortic balloon pump, renal failure requiring dialysis, neurologic deficits (stroke), bleeding requiring exploration, and peri-operative mortality including the intra- and post-operative period.

The following postoperative outcomes were recorded: low cardiac output syndrome (LCOS), which is a postoperative requirement of inotropic agents or intra-aortic balloon pumps; renal failure requiring dialysis; neurologic deficits (stroke); bleeding requiring exploration; and perioperative mortality including the intra- and postoperative periods. Patients’ postoperative length of stay (LOS) in intensive care units (ICU) and postoperative mechanical ventilation times were calculated. Additional perioperative data on arrhythmias were collected from the patients’ medical records. Atrial fibrillation of any duration was confirmed with an electrocardiogram. Perioperative mortality was assumed to include deaths, regardless of cause, that occurred within 30 days after surgery either in or out of hospital.

### 2.5. Operative Technique

All CABG surgeries were performed under general anesthesia with standard median sternotomy. Cardiopulmonary bypass was used in all the surgeries, with cross-clamping of the aorta during cardioplegic arrest and moderate hypothermia. In all patients, multidose cold blood cardioplegia was administered intermittently through the aortic root and retrogradely through the coronary sinus for myocardial protection. CABG was performed using conventional techniques, and complete revascularization was achieved in all patients.

### 2.6. Statistical Analysis

Statistical analyses were performed using the MedCalc Statistical Software Program version 17.2 (MedCalc, Ostend, Belgium). Continuous variables were presented as mean ± standard deviation and categorical variables were presented as the number of subjects as a percentage of the total number of subjects. A comparison of the parametric values between the two groups was made using the Student’s *t*-test or the Mann–Whitney *U*-test, as appropriate. A chi-squared test was used to compare categorical variables between the groups. A multivariate logistical regression analysis was carried out to identify independent predictors for postoperative adverse events. Factors entered into the multivariate model were those with *p* values <0.1 as determined by the univariate analysis. The predictive values of NT-proBNP level and a combination of NT-proBNP and sST2 levels were estimated by comparing the areas under the receiver operating characteristic (ROC) curves. DeLong’s test was used to compare the area under curves (AUC) from each of the models [11]. Moreover, the increased discriminative value after the addition of NT-proBNP level to sST2 level was estimated using the Net Reclassification Improvement (NRI) and Integrated Discrimination Improvement (IDI) [12]. Two-sided *p* values <0.05 were considered statistically significant.

## 3. Results

Patients were divided into two groups according to the sST2 reference value of 35.0 ng/mL: a “high” group with sST2 levels ≥35 ng/mL and a “low” group with sST2 levels <35 ng/mL. The baseline characteristics of both groups are summarized in Table 1. The mean age and sex distributions were similar in both groups (*p* = 0.6078 and *p* = 0.2498 for the high and low sST2 groups, respectively); however, patients in the high sST2 group had significantly higher BMIs (*p* = 0.0329). Patients in the high sST2 group (*n* = 14, 17.5%) had a significantly higher NYHA classification, a higher incidence of previous MI and PCI, and a higher syntax score (*p* = 0.0009, *p* < 0.0001, *p* < 0.0001, and *p* < 0.0001, respectively). Although LVEDD and NT-proBNP levels were significantly higher in patients in the high sST2 group, LVEF was significantly lower in this group (*p* = 0.0436, *p* < 0.0001, and *p* = 0.0192, respectively). The operative data are presented in Table 2. The surgical details were similar for each group and there were no intraoperative deaths.

Clinical outcomes are summarized in Table 3. Postoperative adverse events were more common in the high sST2 group than in the low sST2 group (100% vs 26%, *p* < 0.0001). Two patients died during the postoperative period due to multiple organ failure. Patients in the high sST2 group had a significantly higher mortality rate (*p* = 0.0295), higher need of postoperative inotropy (*p* < 0.0001), higher intra-aortic balloon pump counterpulsation (IABP) placement (*p* = 0.0020), and more frequent respiratory failure (*p* = 0.0295). Patients in the high sST2 group also had longer mechanical ventilation times (*p* < 0.0001) and longer ICU/overall hospital stays (*p* = 0.0041 vs *p* < 0.0001).

Additionally, in the high sST2 group, the incidence of postoperative AF was 23%, and AF was observed significantly more frequently in the high sST2 group than in the low sST2 group (*p* < 0.0001). The univariate logistical regression analysis showed that the following factors were predictors of postoperative adverse events: history of stroke, previous myocardial infarction, previous percutaneous intervention, NYHA class, syntax score, STS score, sST2 level, and NT-proBNP level. The multivariate analysis showed that sST2 level was independently associated with postoperative adverse events (OR: 1.117 (95% CI: 1.016–1.228), *p* = 0.022; Table 4).

The AUC curves of NT-proBNP and sST2 were 0.765 (95% CI: 0.650–0.880, *p* < 0.001; Figure 1) and 0.812 (95% CI: 0.710–0.913, *p* < 0.001), respectively. An NT-pro BNP level of 210 pg/mL was identified as the optimal cut-off value with a sensitivity of 67.7% and a specificity of 67.3%. When sST2 level was added to NT-proBNP level, the area under the AUC curve became 0.822 (95% CI: 0.724–0.919, z = 1.163, difference *p* = 0.2447; Figure 1). Although NT-proBNP, and NT-proBNP plus sST2, had similar accuracies for predicting postoperative adverse events compared to NT-proBNP, sST2 plus NT-proBNP was related to a significant NRI of 46.1% (z = 2.008, *p* = 0.0446) and a significant IDI of 0.0551(z = 2.1855, *p* = 0.0288). An sST2 level of 26.50 ng/mL was identified as the optimal cut-off value with a sensitivity and specificity of 74.1% and 75.3%, respectively. Patients with an sST2 level ≥26.50 ng/mL were significantly more likely to experience postoperative adverse events compared with patients with an sST2 level <26.50 ng/mL (71% vs 21%, *p* < 0.01).

## 4. Discussion

In the present study, we assessed the prognostic value of sST2 in patients with reduced LVEF undergoing CABG. We found that sST2 was an independent predictor of postoperative adverse events in the studied patients. The level of surgical risk was assessed for all patients using preoperative STS scores. Although STS scores were similar in both patient groups, we found that postoperative mortality was significantly more common in the group with sST2 levels ≥35.0 ng/mL, and that elevated preoperative sST2 levels were significantly associated with an increased risk of death. We also found that patients with high levels of preoperative sST2 had a higher incidence of postoperative low cardiac output syndrome (as a result of requiring inotropy and IABP), respiratory failure, and longer postoperative hospital stays.

sST2 is a novel biomarker that has been shown to predict adverse outcomes and death in patients with heart failure and MI [13]. In the HF-ACTION heart failure study, ST2 levels were found to be a significant predictor of mortality due to cardiovascular disease and hospitalization, with a cut-off level of 35 ng/mL. Risk of mortality was also found to be significantly higher in patients with ST2 levels ≥35 ng/mL [14]. In the Framingham Heart Study, sST2 concentrations were shown to be related to systolic blood pressure and diabetes. That study also found that male gender and old age were related to higher sST2 concentrations by adjusting for coronary artery disease, heart failure, atrial fibrillation, diabetes, hypertension, obesity, valvular disease, left ventricular systolic dysfunction, and pulmonary and renal dysfunction [15]. In the present study, no associations were found between sST2 and DM, HT, age, or gender. The significance of sST2 in CABG surgery was first evaluated by Lobdell et al. [8], who found that preoperative sST2 levels were associated with postoperative acute kidney injury. However, in the present study, no association was observed between sST2 levels and acute renal failure. The predictive ability of sST2 and NT-proBNP has been studied in patients undergoing cardiac surgery, and both markers have been shown to improve a model classification of in-hospital mortality [9].

Ramkumar et al. found that elevated pre- and post-operative levels of sST-2 or NT-proBNP are associated with lower survival rates after cardiac surgery, and the authors claimed that both biomarkers can be used to assess postoperative prognosis [16]. Additionally, Brown et al. showed that some cardiac biomarkers such as sST-2, NT-proBNP, galectin-3, and cystatin-C significantly improved the prediction of readmission or mortality within 30 days after coronary artery bypass graft surgery [17]. In our study, we found that elevated preoperative sST2 levels were associated with an increased incidence of postoperative adverse events, including arrhythmia, respiratory failure, and inotropic and IABP requirement. Furthermore, we found that mechanical ventilation times and length of hospital stay were longer in patients with high levels of sST2.

As well as sST2, NT-proBNP is another powerful biomarker for the diagnosis and prognosis of heart failure. Levels of this biomarker have been found to be elevated in cases of ventricular wall stress [18]. In the present analysis, patients with elevated sST2 levels were found to have significantly increased NT-proBNP levels, and reduced levels of LVEF and sST2 were found to be positively correlated with NT-proBNP levels. Many traditional risk factors have been identified for AF following CABG, such as advanced age, obesity, and low ejection fractions [19,20,21,22,23]. Several cardiac biomarkers have also been examined for peri-operative atrial fibrillation and flutter (POAF) after CABG, such as troponin, CK-MB, and NT-proBNP. Gasparovic et al. [24] examined the impact of NT-proBNP levels on the occurrence of AF in patients undergoing CABG surgery and showed that elevated preoperative NT-proBNP levels are a strong predictor of AF. Our study showed that sST2 levels are a more sensitive and specific predictor than these aforementioned traditional risk factors and biomarkers.

At present, the most widely used surgical risk scoring systems, such as EuroSCORE and STS, do not include a specific cardiac biomarker variable for evaluating adverse outcomes. In this study, we found that levels of sST2, a novel cardiac biomarker primarily used to predict survival after heart failure and myocardial infarction, were associated with adverse outcomes in patients with impaired LV function undergoing CABG surgery. Coronary bypass surgery is commonly more complicated in patients with an impaired left ventricle. According to our results, preoperative sST2 levels may be used as a biomarker to predict postoperative outcomes in this population; however, more prospective and multicenter studies are needed to support our findings.

Interestingly, our research also showed that elevated preoperative sST2 levels were associated with an increased risk of postoperative AF. An association between sST2 and cardiac arrhythmia was first described by Okar et al. [25], who found that sST2 levels were an independent predictor of recurrent atrial fibrillation after cryoablation. Moreover, another study demonstrated that levels of circulating sST2 were independently associated with a history of ventricular arrhythmia in patients with right ventricular failure [26]. The higher risk of developing postoperative atrial fibrillation may result from an increase in the extent of myocyte necrosis and myocardial fibro-fatty replacement due to ventricular dysfunction.

The present study has several limitations. Firstly, it was a single-center study with a small sample size and a predominantly male patient population. Additionally, the data were collected from our surgical practice, and therefore our patient population, surgical techniques, and postoperative management may have affected the outcome. Furthermore, we had no opportunity to include certain factors—such as fragility, inflammation, or neurohumoral activation—which may have influenced postoperative outcomes. Another limitation is that our results are based on a single sST2 measurement per patient; serial measurements of this biomarker may provide greater prognostic value. Our results need to be confirmed by large multi-center prospective trials.

## 5. Conclusions

Elevated preoperative sST2 levels were found to be associated with a higher risk of postoperative adverse events in patients with impaired left ventricular function undergoing CABG surgery. Soluble ST2 may be a useful cardiac biomarker to predict adverse postoperative adverse events in such patients.

## Figures and Tables

**Figure 1 medicina-55-00572-f001:**
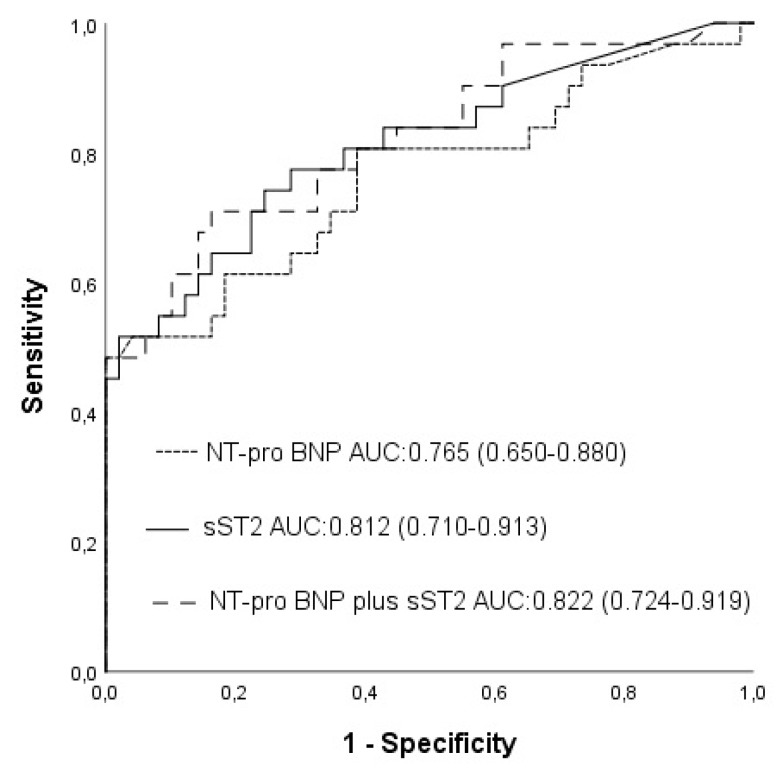
Receiver operating characteristic (ROC) curves for soluble ST2 (sST2), N-terminal pro-brain natriuretic peptide (NT-proBNP), and NT-proBNP plus sST2, which were used for predicting postoperative adverse events.

**Table 1 medicina-55-00572-t001:** Preoperative patient characteristics.

Variables	Patients with ST2 <35 (*n* = 66)	Patients with ST2 ≥35 (*n* = 14)	*p* Value
Age (y)	63.2 ± 9.3	62.4 ± 7.3	*p* = 0.6078
Gender			
Male	52 (78%)	9 (64%)	*p* = 0.2498
Female	14 (21%)	5 (35%)	*p* = 0.247
BMI (kg/m^2^)	26.7 ± 3.9	29.2 ± 3.8	*p* = 0.0329
Diabetes	30 (45%)	6 (42%)	*p* = 0.8600
Hypertension	12 (18%)	2 (14%)	*p* = 0.7291
Hyperlipidemia	44 (66%)	8 (57%)	*p* = 0.5001
Pulmonary disease	11 (16%)	4 (28%)	*p* = 0.3030
Smoking	39 (59%)	7 (50%)	*p* = 0.5346
Previous MI	6 (9%)	11 (78%)	*p* < 0.0001
Previous PCI	14 (21%)	11 (78%)	*p* < 0.0001
PAD	9 (13%)	4 (28%)	*p* = 0.1715
LVEF (%)	42.4 ± 3.5	39.6 ± 5.3	*p* = 0.0192
LVEDD (mm)	49 (41–61)	53 (44–64)	*p* = 0.0436
LVESD (mm)	33 (28–40)	34 (30–56)	*p* = 0.1753
NYHA Class			*p* = 0.0009
I n (%)	32 (49)	2 (14)	
II n (%)	33 (50)	7 (50)	
III n (%)	1 (2)	5 (36)	
CAD vessel no.			*p* = 0.1189
2 vessel	5 (7%)	3 (21%)	
3 vessel	61 (98%)	11 (78%)	
LMCA disease	13 (19%)	2 (14%)	*p* = 0.6396
STS score *	1 (0.3–4.2)	0.8 (0.5–5.9)	*p* = 0.3739
Syntax score *	22 (11–45.5)	34 (32–38)	*p* < 0.0001
NT pro-BNP *(pg/mL)	160 (44–455)	1680 (750–8130)	*p* < 0.0001
Creatinin (mg/dL)	0.83 ± 0.23	0.91 ± 0.24	*p* = 0.2175
BUN (mg/dL)	38.1 ± 12.7	39.3 ± 9.3	*p* = 0.3649

Abbreviations: BMI: body mass index, BUN: blood urea nitrogen, CAD: coronary artery disease, LMCA: left main coronary artery, LVEDD: left ventricular end-diastolic diameter, LVEF: left ventricular ejection fraction, LVESD: left ventricular end-systolic diameter, MI: myocardial infarction, NYHA: New York heart association, NT-pro BNP: N-Terminal pro-brain natriuretic peptide, PAD: peripheral arterial disease, PCI: percutaneous coronary intervention, STS: society of thoracic surgeon.* Comparison was made using Mann–Whitney *U* test at *p* < 0.05, and these values were described by median with inter-quartile range (25th and 75th percentile).

**Table 2 medicina-55-00572-t002:** Operative characteristics.

Variables	Patients with ST2 <35 (*n* = 66)	Patients with ST2 ≥35 (*n* = 14)	*p* Value
Bypass vessel no. (%)			*p* = 0.4138
2	3 (4)	2 (14)	
3	26 (39)	5 (35)	
4	32 (48)	7 (50)	
5	5 (7)	0 (0)	
CPB time, min	95 ± 18	89 ± 19	*p* = 0.4980
X-Clamp time, min	51 ± 11	57 ± 13	*p* = 0.0574
Use of LIMA *n* (%)	63 (95)	14 (100)	*p* = 0.4191
Endarterectomyn (%)	7 (10)	2 (14)	*p* = 0.6941

Abbreviations: CPB: cardio pulmonary bypass, LIMA: left internal mammary artery.

**Table 3 medicina-55-00572-t003:** Postoperative adverse events.

Variables	Patients with ST2 <35 (*n* = 66)	Patients with ST2 ≥35 (*n* = 14)	*p* Value
Operative mortality *n* (%)	0	0	-
Postoperative mortality *n* (%)	0	2 (14%)	*p* = 0.02953
Need of inotrophy *n* (%)	6 (9%)	12 (85%)	*p* < 0.0001
Need of IABP *n* (%)	1 (1%)	3 (21%)	*p* = 0.0020
MV time, hour	8 (5–28)	18 (7–72)	*p* < 0.0001
Stroke *n* (%)	0	0	-
Renal failure requiring dialysis *n* (%)	2 (3%)	2 (14%)	*p* = 0.0811
Re-operation for bleeding *n* (%)	3 (4%)	1 (7%)	*p* = 0.6873
Respiratory failure *n* (%)	0	2 (14%)	*p* = 0.02953
Arrhythmia *n* (%)	8 (12%)	11 (78%)	*p* < 0.0001
ICU length of stay, days *	2.5 (1–6)	4 (2–13)	*p* = 0.0041
Overall length of stay, days *	8 (5–17)	11 (8–23)	*p* < 0.0001
Postoperative adverse events *n* (%)	17 (26)	14 (100)	*p* < 0.0001

Abbreviations: IABP: intra aortic balloon pump, ICU: intensive care unit, MV: mechanic ventilation. * Comparison was made using Mann–Whitney *U* test at *p* < 0.05, and these values were described by median with inter-quartile range (25th and 75th percentile).

**Table 4 medicina-55-00572-t004:** Univariate and multivariate logistic regression analysis for prediction of postoperative adverse events.

Variable	Univariate			Multivariate		
	OR	95% CI	*p* Value	OR	95% CI	*p* Value
Previous MI	8.125	2.335–28.270	0.001			
History of stroke	7.111	0.756–66.894	0.086			
Previous PCI	4.741	1.728–13.009	0.003			
NYHA ≥ 2	3.250	1.219–8.666	0.019			
Syntax score	1.143	1.056–1.237	0.001			
STS score *	1.682	0.962–2.941	0.068			
sST2 levels	1.128	1.065–1.194	<0.001	1.117	1.016–1.228	0.022
NT-proBNP levels	1.004	1.001–1.007	0.004			

Abbreviations: CI: confidence interval, OR: odds ratio, **MI:** myocardial infarction, NT-proBNP: N-terminal pro-brain natriuretic peptide, NYHA: New York heart association, **PCI:** percutaneous coronary intervention, sST2: soluble ST2, STS: society of thoracic surgeon. * This parameter is not entered to the model in order to prevent multicollinearity.

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
