# Peer review of "Using Soluble ST2 to Predict Adverse Postoperative Outcomes in Patients with Impaired Left Ventricular Function Undergoing Coronary Bypass Surgery"

_medicina, 2019, doi:10.3390/medicina55090572_

Round 1

Reviewer 1 Report

Introduction: Scoring systems are not in the focus of this study (line 64-66). This part should be deleted.  The end of the introduction should provide a hypothesis which is either proven or not by the authors.

Methods

How many blood samples were measured for sST2 from any patient? This is especially important for the gray area of measurement around 35ng/ml since the group >35ng/ml is very small.

Discussion. This section should be focused on the results of the study. A critical appraisal regarding the small sample size should be included. The authors should include some sentences about consequences of the results for daily clinical practice.  

Statistics: The statistic section should be amended with the definition of a primary endpoint, the secondary endpoints and most of all a power calculation of the sample size. A clinical relevant difference between both groups should be defined. This especially relevant, since the second group is very small and the authors analyzed many clinical endpoints.

Minor typos should be corrected: e.g. line 75 “An eighty” or line 80 “rthym”

In the presented for a publication in medicina cannot be supported. Relevant improvements are required prior to publication.

Reviewer 2 Report

The Authors perfromed a novel study which was dealing with the prognostic value od pre-operative sST2 in post-CABG patients with LV dysfunction. I have one, yet important comment: Although the paper describes novel findings, the Authors have not performed multivariate analysis. Admittedly, their ROC analysis demonstrated that sST2 was more  sensitive and sensitive than other predictors, nevertheless they should investigate whether sST2 provides any additional prognostic value when added into the traditional predictors, especially NT-proBNP.

Reviewer 3 Report

1.      The title should be reconsidered and written correctly

2.      Please explain how it may be, that primary endpoint is association?

Row83 “The primary endpoint of our study was association between pre-operative soluble ST2 level and post-operative outcomes.”

3.      Groups of patients are unequal. What could be an impact to your results?

4.      Row 133: median (min and max) could be changed into median [interquartile range]

5.      Row 141or 142 should be deleted

6.      Row 147 The word “patient” should be in small letter

7.      Please rewrite Univariable analysis using Cox model explanations, because this analysis doesn't show correlation.

8.      Table 1:

* why there is no female p value?

*  Please explain how BMI could be linked to elevated soluble St2?

* What was the period between previous MI, PCI and soluble St2 eveluation before CABG? May that period implement results?

* CAD 3 vessel – there is no p value meaning

9.      Table 1 footnotes: should be written values expression of variables (mean+- SD etc.)

10.  Table 2:

* How will you explain, that X-Clamp time is shorter in ST2>35 group of patients?

*Table 2 footnotes: should be written values expression of variables (mean+- SD etc.)

11.  Table 3:

Table 3 footnotes: should be written values expression of variables (mean+- SD etc.)

12.  Please rewrite Univariate regression analysis Table 4 – it is not clear what you want to say

13.  Figure 1 – there is no footnotes, please write

14.  Discussion:

* it is very brave to say, that you showed the sST2 effect on mortality, making the conclusion from 2 deaths. What was the OR in post-operative mortality based on univariate analysis using ROC test?

* there are written examples what was found in other studies and in this stud, but there are no explanation why there are differences.

15.  Conclusion: I would be sure for conclusion based on 2 cases of death.

Reviewer 4 Report

1. The patient numbers is limited and only 14 in Patients with ST2 ≥ 35

2. Patients with ST2 ≥ 35 had significantly higher previous MI, previous PCI,  syntax score and NT pro BNP. In addition, significant poor LVEF and NYHA Fc class were also revealed in Patients with ST2 ≥ 35. The result of poor  postoperative outcomes could predicted in these traditional baseline disease and risk factors. I don't think ST2 ≥ 35 can be a independent value to predict the poor  postoperative outcomes.

3. I suggest authors to collect more patients and showed the relationship of absolute soluble St2 value and clinical outcomes.

Round 2

Reviewer 2 Report

The Authors have improved their manuscript in accordance with the Reviewers' comments. Admittedly, they were not able to increase the sample size, nevertheless, their study was based on a low-volume center, which was mentioned in their reply.

Reviewer 3 Report

P { margin-bottom: 0.08in; }

English language corrections are needed.

Results: |The area under the curve (AUC) of the NT-proBNP was 0.765 (95% CI: 0.650–0.880, p < 0.001). When ST2 was added to NT-proBNP became 0.822 (95% CI: 0.724–0.919, z = 1.163, difference p = 0.2447, figure 1)).” What was the results of ST2?

Results: “A ST2 value of 26.50 was identified as the optimal cut-off value with sensitivity and specificity of 74.1% and 75.3% , respectively” So how it increase the adverse outcomes?

Table 4: What are variables cut off? If you use ROC test, you should use OR insted of HR. HR:1.117 (95% CI:1.016-1.228, p = 0.022)  is very low.

Figure 1: if the main variable in this article is ST2, so why you don't show ST2 ROC curve?

Reviewer 4 Report

Thanks for your adequate revision. 
